# Chain-of-region: Visual Language Models Need Details for Diagram Analysis

**Xue Li**[1][*]**, Yiyou Sun**[2][*]**, Wei Cheng**[3]**, Yinglun Zhu**[4]**, Haifeng Chen**[3]
[1]University of Wisconsin-Madison, [2]University of California, Berkeley, [3]NEC Laboratories America
[4]University of California, Riverside

## Abstract

Visual Language Models (VLMs) like GPT-4V have broadened the scope of LLM applications, yet they face significant challenges in accurately processing visual details, particularly in scientific diagrams. This paper explores the necessity of meticulous visual detail collection and region decomposition for enhancing the performance of VLMs in scientific diagram analysis. We propose a novel approach that combines traditional computer vision techniques with VLMs to systematically decompose diagrams into discernible visual elements and aggregate essential metadata. Our method employs techniques in *OpenCV* library to identify and label regions, followed by a refinement process using shape detection and region merging algorithms, which are particularly suited to the structured nature of scientific diagrams. This strategy not only improves the granularity and accuracy of visual information processing but also extends the capabilities of VLMs beyond their current limitations. We validate our approach through a series of experiments that demonstrate enhanced performance in diagram analysis tasks, setting a new standard for integrating visual and language processing in a multimodal context.

## 1 Introduction

Recent advances in Visual Language Models (VLM), such as GPT-4V (OpenAI, 2023) and Gemini (Team et al., 2023), have demonstrated broad applicability in multimodal downstream tasks, including navigation (Yan et al., 2023), search (Cloud, 2024), and answering scientific questions (Yue et al., 2024b). Despite these achievements, significant limitations persist in the perceptual capabilities of VLMs. They often fail to discern fine visual details (Tong et al., 2024; Peng et al., 2024), are easily misled by visual distractors (Zhang et al., 2024b), struggle with understanding visual relationships (Yue et al. (2024b), Fig. 43), and may even hallucinate non-existent objects (Huang et al., 2024). A notable example is shown in Figure 1 (a), where GPT-4 cannot accurately determine the specific value from a bar graph. Such perceptual flaws substantially impede the ability of LLMs to perform rigorous scientific analysis on diagrams.

Addressing these perceptual issues raises questions about the capacity limit of VLMs. It remains uncertain how much visual instruction data is necessary (Zhang et al., 2024a; Liu et al., 2024b) or whether it is feasible to develop a VLM that can impeccably manage visual details in complex diagrams given computational constraints (Bai et al., 2024). A promising approach to overcoming these limitations involves a sequential inference of visual elements, rather than generating responses in a single pass. This iterative strategy is previously validated in works like Tree-of-thought (Yao et al., 2024). Unlike tasks expressed in natural language, visual tasks often necessitate further decomposition into specific regions. This paper focuses on the following critical research question:

*How can regions be decomposed and visual details collated for scientific diagram analysis?*

One straightforward approach involves using segmentation methods such as SAM (Kirillov et al., 2023) to generate regions of interest with a segmentation mask. This strategy, used in recent studies (Yang et al., 2023; Yan et al., 2024), tags each object with a number along with the segmentation mask before VLM analysis. However, when applied to scientific diagrams, the accuracy of segmentation

---

[*]Equal Contribution

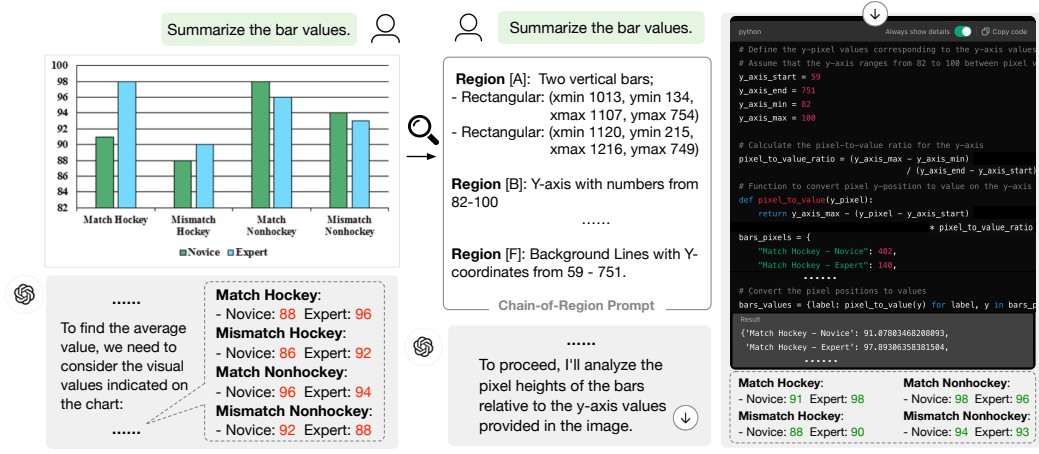

Figure 1: Comparison of GPT-4o (OpenAI, 2023) and the one augmented with a Chain-of-Region (CoR) prompt in addressing queries about scientific diagrams. (a) illustrates the difficulty GPT-4o encounters when directly inferring exact values from a bar graph. (b) demonstrates how additional visual details (such as pixel location of background lines and rectangular parameters) along with the original figure can be further utilized to accurately infer the accurate bar values.

methods is limited by several factors: a) generalization issues, as methods trained on natural images often fail with out-of-distribution input such as scientific diagrams (details in the Appendix B); b) difficulty in detecting small or thin objects, such as lines or dots in charts; and c) the inability to infer hidden shape parameters, such as the rotation angle of a rectangle or the slope of a line.

To overcome these challenges, we propose a novel integration of nearly forgotten yet potent traditional computer vision techniques (commonly implemented in OpenCV (Bradski, 2000)) with VLMs to *initialize* (Section 3.1.1), *split* (Section 3.1.2) and *merge* (Section 3.1.3) regions, and *collect meta-information* (Section 3.2). The effectiveness of traditional techniques hinges on a salient observation of scientific diagrams: unlike natural images, these diagrams have homogeneous colors and structured patterns for each visual element, as they are human-created by software or programs. For example, we flexibly utilize OpenCV operators, such as `cv2.connectedComponents` for initial region separation and `cv2.findContours` to generate contours for shape detection.

We propose the Chain-of-Region (CoR) framework for diagram analysis, which first integrates traditional CV techniques to decompose the diagram to collect visual details and then incorporates these visual details into VLM for a fine-grained analysis (e.g., Figure 1(b)). Our CoR framework has the following compelling advantages:

1. **White-box Algorithm**: Unlike deep learning models that operate as black boxes, the CoR framework employs transparent, rule-based region separation methods. This white-box approach allows for greater interpretability and easier debugging, enabling users to understand and modify the processing steps based on specific requirements of the task.

2. **Cost-Effective**: Compared to DL-based methods, the CoR framework is simpler and more cost-effective. It requires *only CPU* processing, and its operations, which are less computationally intensive, can be executed in milliseconds.

3. **Plug-and-Play Compatibility**: The CoR framework is designed for seamless integration, which can be effortlessly incorporated with any pre-trained VLM. This plug-and-play compatibility ensures that users can leverage existing VLMs without extensive retraining or adaptation.

4. **Strong Empirical Performance**: In the realm of scientific diagram analysis, the CoR framework demonstrates superior performance. Extensive evaluation benchmarks show that it not only enhances the accuracy of visual detail extraction but also improves the overall reliability of the analyses conducted by VLMs. These benchmarks indicate that CoR significantly outperforms existing DL-based methods in terms of precision and recall in complex diagram interpretations.

## 2 RELATED WORK

**Augmented Prompting for VLMs.** VLMs (Dosovitskiy et al., 2021; Zhu et al., 2023; OpenAI, 2023; Liu et al., 2024b;a; Li et al., 2024a) enhance LLMs by integrating visual perception capabilities. To improve their handling of complex images, various prompting techniques have been developed. Iterative strategies, such as those seen in Tree-of-thoughts (Yao et al., 2024), employ visual tags (Yang et al., 2023; Nasiriany et al., 2024), Hierarchical Layout Tree(Fan et al., 2024b), Language Model-guided Diagram Parsing Tools (Wang et al., 2024) to guide VLMs in decision-making processes. Additionally, some researchers employ external tools to enhance VLM prompts. Yang et al. (2023) introduces Set-of-Mark (SoM) Prompting, utilizing segmentation masks to tag regions, while Wu et al. (2024) develops a toolkit that focuses VLM attention on specific regional details with aids like rulers and compasses. Lu et al. (2024) streamlines the integration of image caption tools and text detectors into visual prompting, highlighting the ongoing innovation in VLM applications. Our method integrates traditional CV techniques with VLMs to enhance prompting capabilities further.

**Enhancing Spatial Understanding in VLMs.** Recent studies in enhancing VLMs explore various modifications to improve spatial reasoning and detail recognition. Some research focuses on altering the existing architecture to incorporate spatial input like bounding box (Liu et al., 2024c), learnable spatial prompt Dorkenwald et al. (2024), segmentation masks (Guo et al., 2024), 3D voxels (Man et al., 2024; Chen et al., 2024), depth information (Cheng et al., 2024) and RGBA region-text pairs (Sun et al., 2024). On the other hand, several studies aim to augment VLMs by using additional tuning data Yan et al. (2024); Liu et al. (2024b;a); Xuan et al. (2024); Sun et al. (2024); Chen et al. (2024); Li et al. (2024b). This approach is evident in the work by Zhang et al. (2024c); Meng et al. (2024); Han et al. (2023); Fan et al. (2024a) who gather visual instruction data specifically on charts and fine-tune VLMs for better performance in interpreting such data. While these approaches necessitate fine-tuning procedures, heavy human effort in dataset preparation, and often face generalization challenges, our method integrates traditional CV techniques with VLMs to more efficiently handle complex scientific diagrams.

## 3 METHODOLOGY

This section outlines the Chain-of-Region pipeline, as it reflects the sequential processing of separate visual regions to form a cohesive analysis chain, and its application in enhancing the perceptual abilities of VLMs for handling complex scientific diagrams.

**Overview.** Typically, a VLM denoted as $F$, takes an image $X_{im} \in \mathbb{R}^{H \times W \times 3}$ and a text query $X_q$, and generates a textual output $\hat{Y} \sim F(X_{im}, X_q)$ aimed to closely match the desired response $Y$. However, it is counterintuitive, even for humans, to answer complex questions, such as the one in Figure 1, with just a single glance (or one function pass). This often requires a detailed visual assessment of each component represented in a diagram, such as reading bar values iteratively.

To mimic this methodical process, we employ a divide-and-conquer approach that iteratively extracts and compiles detailed region-based information into the VLM's prompt. As demonstrated in Figure 2, this process consists of two primary steps: (a) **Region Identification** (Section 3.1): This step involves designing an algorithm to identify distinct regions within the image. These regions correspond to discrete visual elements or clusters of elements relevant to the text query. (b) **Information Assembly** (Section 3.2): Once the regions are identified, this step focuses on extracting specific information from each region. The extracted data are then systematically aggregated and integrated into the VLM prompt, enhancing the model's ability to generate accurate and contextually relevant responses.

### 3.1 REGION IDENTIFICATION

At a high level, the Region Identification process includes three stages: initialization, splitting, and merging, as illustrated in Figure 2. The pipeline begins with **Region Initialization** using the `cv2.connectedComponents` method to identify initial regions (Section 3.1.1), demonstrated by separating a graph into 16 distinct regions in Figure 2(a). However, certain visual elements, such as lines and circles, may remain unseparated due to being connected components, posing challenges in isolating individual parameters. These regions then undergo **Region Splitting** (Section 3.1.2), where both structured and unstructured splits are performed. During this stage, critical parameters such as

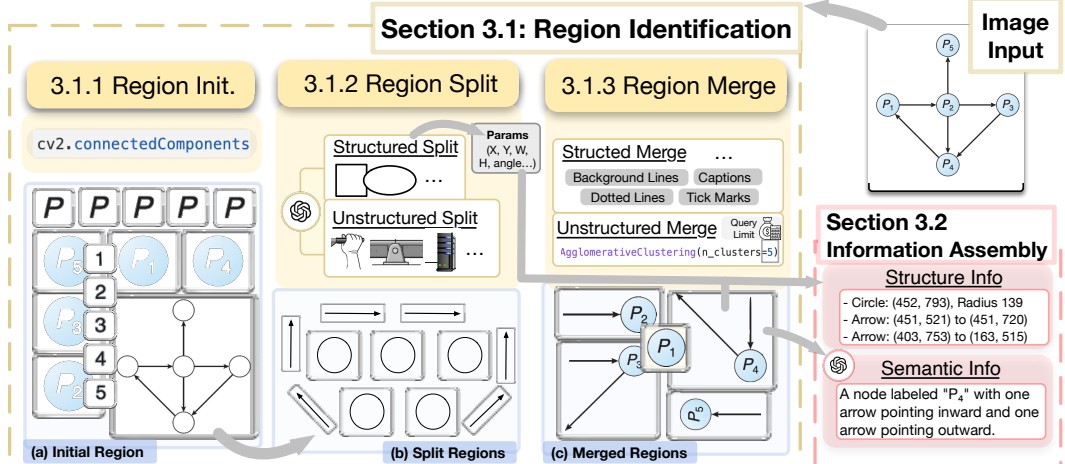

Figure 2: Overview of the Chain-of-Region (CoR) framework. Each gray-boundary box represents an individual region. In (a), the initial regions are identified. Moving from (a) to (b), one large region is split into circles and arrows, using structured splitting. From (b) to (c), all existing regions are merged into distinct clusters. In red box, we exemplify the information assembly process with a merged region containing `P4` node.

pixel locations, width, height, and angles are extracted for further analysis. As our strategy focuses on sequentially accessing regions to gather detailed information, we must consider the budget limit on the number of regions that can be sent to the VLM for inference. To address this, in **Region Merging** (Section 3.1.3), regions are merged by considering various structural elements like background lines and captions. These regions are further clustered by `AgglomerativeClustering` to ensure that the number of regions follows the budget limit.

### 3.1.1 REGION INITIALIZATION

At the first stage, we initialize the region using connected components, based on the observation that visual elements in scientific diagrams exhibit homogeneous colors and structured patterns. This process begins with the conversion of the input $X_{im}$ into a binary image $X_{bi}$, facilitating the separation of distinct visual components. The code snippet for this process is as follows:

```
1 _, X_bi = cv2.threshold(X_im, 0, 1, cv2.THRESH_OTSU)
2 _, X_fg = cv2.connectedComponents(X_bi)
3 _, X_bg = cv2.connectedComponents(1 - X_bi)
4 X_region = X_fg + X_bg + (X_bg > 0) * offset
```

First, the image is converted into grayscale and then into a binary format using a thresholding technique. The `cv2.threshold` function applies Otsu's thresholding method (Otsu et al., 1975), which automatically determines the optimal threshold value for separating the image into foreground and background. The resulting binary image $X_{bi}$ contains clear distinctions between the visual elements and the background.

Next, we utilize the `cv2.connectedComponents` function to identify and label the connected regions within $X_{bi}$. The connected components algorithm assigns a unique label to each connected region based on 8-connectivity[1]. The output, $X_{fg}$, represents the initial segmentation of the diagram where each unique label in the pixel location corresponds to a different visual component.

Additionally, we process the inverse of the binary image, $1 - X_{bi}$, to capture any components that might have been categorized as background in the original binary segmentation. This step ensures that regions which are visually connected but appear as dark on a light background are also identified as distinct entities. The result, $X_{bg}$, provides a complementary set of region masks.

The final step involves combining these $X_{fg}$ and $X_{bg}$ to create a unified region map, $X_{reg}$. The term $(X_{bg} > 0) * \text{offset}$ ensures that the labels from $X_{bg}$ are adjusted by a specified offset that guarantees

---

[1]8-connectivity refers to the concept in image processing where a pixel is considered connected to its eight surrounding pixels, including diagonally adjacent ones.

that there are no overlapping region labels, since both region maps begin the labeling from zero. Together, these steps form a strategy for segmenting scientific diagrams into distinct and analyzable regions, setting the stage for further processing steps involving region splitting and merging.

### 3.1.2 REGION SPLIT

The initial region identification described provides a basic segmentation of visual elements. However, as illustrated in Figure 2, this method may not adequately separate connected objects with overlapping or connected elements. To address this limitation, we introduce two approaches: structured split and unstructured split.

**General Split Procedure.** We first introduce the general region-splitting procedure. The following Python function outlines the core of our splitting process:

```python
def split_regions(M, M_sub):
    M_remain = cv2.subtract(M, M_sub)
    _, X_remain = cv2.connectedComponents(M_remain)
    X_region = update_region(X_region, X_remain)
```

This function takes two masks: $M$, the original mask of the region, and $M_{sub}$, a mask representing a subset of $M$ that needs to be separated. Using OpenCV's subtract method, we remove $M_{sub}$ from $M$ to isolate the remaining parts of the region. We then apply connectedComponents to identify newly isolated components within the remaining mask, $M_{remain}$. These components are integrated into the existing region map $X_{region}$ through the update_region function, which effectively updates our overall region separation by incorporating the new distinct regions.

**VLM-Assisted Structure Recognition.** Before proceeding with specific region splitting, we employ a VLM to preliminarily discern the potential structures or shapes within each main region, prioritizing regions based on their area sizes. This discernment step, detailed further in the Appendix A.1, involves a prompt-based query to the VLM which suggests possible structural categories. The VLM's response then directly triggers the corresponding structure or shape detector. If the VLM does not recognize the structure as one present in our current library, the process defaults to an unstructured component split. Note that this discerning step is constrained to several main regions with large area size and limited by a pre-defined recognition call limits (ex. 10) per diagram.

**Structured Split.** For regions identified as containing pre-defined structures, the splitting process is specialized to handle standard forms, such as rectangles, ellipses, and lines. We employ a structure detector that processes each region independently based on its labeled mask ($M_i \in \mathbb{R}^{W \times H}$) [2]:

```python
M_shape, S_i = structure_detector(M_i)   # S_i: the shape parameters
split_regions(M_i, M_shape)
```

The function structure_detector analyzes $M_i$ and identifies sub-regions $M_{shape}$ that correspond to distinct structured elements within the region, along with their parameters $S_i$. The detected sub-regions are then split from the main region using the described split_regions function.

We include a set of common structures typically encountered in the analysis of scientific diagrams (detailed in Appendix A.2). This includes both third-party algorithms such as the ellipse detector (Meng et al., 2020) and the segment detector (Von Gioi et al., 2008), as well as our own implementations for detecting rectangular shapes. These structure detectors enable the extraction of accurate meta-information, such as the pixel location of rectangles, which are critical for detailed diagram analysis.

**Unstructured split.** Dealing with unstructured shapes presents a distinct challenge compared to structured shapes, as it is impractical to design a "universal" computer vision algorithm that accommodates all visual elements effectively. Our unstructured split algorithm stems a straightforward intuition: composite visual elements can often be deconstructed into their constituent parts based on "connectivity". For instance, consider the example depicted in Figure 3 (a), where a cup shape shows a weakly connected spot at the juncture between the bottle and the stick. This intuition underpins our methodology.

---

[2]It can be obtained by comparing the region map $X_{region}$ against the label $i$: $X_{region} == i$.

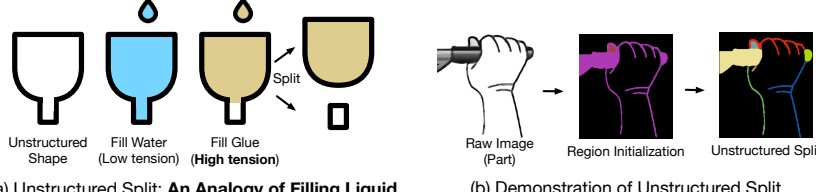

Figure 3: Illustration of the unstructured split principle. (a) The splitting process is analogous to filling a high-tension liquid into a container, where the shape is divided into filled (yellow) and unfilled (white) regions. (b) A real-world example demonstrating the separation of an unstructured shape "handle" from a hand. Different colors represent different separated regions. )

Our algorithm can be viewed as the process of filling a high-tension liquid within the composite shape, thereby identifying natural divisions between the filled and unfilled regions. A simple implementation of this concept can simulate this filling and splitting process:

```
1  T = cv2.distanceTransform(M_i, cv2.DIST_L2).max() // 2 # T: tension force
2  kernel = cv2.getStructuringElement(cv2.MORPH_ELLIPSE, (T, T))
3  M_shape = cv2.morphologyEx(M_i, cv2.MORPH_OPEN, kernel, iterations=1)
4  split_regions(M_i, M_shape)
```

This code snippet outlines our algorithm's operation. The `cv2.distanceTransform` function calculates the distance to the nearest zero pixel for each pixel of the binary image $M_i$ (the mask of the current region). The maximum distance found is halved to define a tension threshold $T$, which serves as a basis for the ellipse-shaped kernel. This kernel is used in `cv2.morphologyEx` with `cv2.MORPH_OPEN`, which separates the image based on the defined tension areas. The morphological opening helps to isolate regions within the image by breaking down weak connections, as simulated by our high-tension liquid analogy. This unstructured split technique is advantageous for analyzing scientific diagrams where shapes may not conform to standard geometric norms but still exhibit distinct connectivity properties that can be exploited for detailed analysis.

### 3.1.3 REGION MERGE

After splitting, we often encounter scenarios where hundreds of separated regions have been identified. Given budget constraints, it is impractical to extract information by querying the VLMs for each region individually. Therefore, in this section, we discuss methods to reduce the number of regions by merging several smaller regions into more comprehensive ones.

**Structured Merge.** The first approach to region merging is structured, relying on both existing tools and heuristic rules. (a) We utilize mature OCR tools (Contributors, 2024) to detect and generate caption boxes within the diagram. Regions that are enclosed within these text boxes are considered related and are merged into a single region; (b) We apply a set of heuristic rules to merge regions based on visual patterns that follow fixed rules, such as dotted lines and background lines (details in Appendix A.3).

**Unstructured Merge.** After the structured merging, we also employ an unstructured approach to further consolidate regions, especially when dealing with complex diagrams where structured methods may not suffice. Specifically, we apply hierarchical clustering (Nielsen & Nielsen, 2016) to the region masks. The clustering process is governed by a distance matrix $D$, where each element $D_{ij}$ represents the distance calculated based on the centroids of the region masks $M_i$ and $M_j$. The hierarchical clustering process will stop when the number of merged regions reaches a predefined budget $B$. This budget acts as a threshold to balance the number of queries made to the VLM against the need to maintain a comprehensive understanding of the diagram. As a result, the final output set of region masks will be $\{M_i^*\}_{i=1}^B$, where each $M_i^*$ represents a merged region ready for further analysis with the VLM.

### 3.2 INFORMATION COLLECTION

The final phase focuses on the collection of meta-information from each region $M_i^*$:

**Structure Information Collection.** First, we collect shape information $S_i$ for each region $M_i^*$, which has been previously collected in the structured split phase. This information includes details about the geometric properties of the region such as contours, area, and perimeter.

**Semantic Information Extraction.** Next, we extract semantic information that includes the role and entity type of each region, along with a detailed description. To facilitate this, we visually highlight the region in image $X_{im}$ masked with $M_i^*$ and fading other regions. It ensures that the VLM's attention is directed towards the relevant region during the analysis. The process for this visualization and the subsequent query construction is detailed in the Appendix A.4. The highlighted image is then sent to the VLM using a specific prompt designed to extract detailed semantic information.

**Aggregation and Query Formulation.** The collected shape and semantic information for each region is aggregated and appended to a master query $X_q$. This query, enriched with detailed regional metadata, is subsequently used to generate a response $\hat{Y}$ from the VLM. The process is illustrated in Figure 8 in Appendix, which shows how each piece of collected information contributes to the formulation of the query and influences the VLM's output.

By systematically collecting and integrating both shape and semantic information, we let VLM not only recognize the physical attributes of the diagram components but also understand their contextual and functional utility.

## 4 EXPERIMENTS

**Datasets.** We conducted extensive experiments on the Massive Multi-discipline Multimodal (MMMU, Yue et al. (2024a)) dataset, which includes a diverse array of multimodal questions sourced from college exams, quizzes, and textbooks. These questions combine image inputs with textual inquiries. While our network is specifically tailored for scientific diagrams, MMMU encompasses a broad range of natural images across categories such as Art & Design. To align our research focus, we conduct experiments on several sub-categories of MMMU that consists exclusively of scientific diagrams. Specifically, this subset includes images from two main categories: Science (encompassing `Biology`, `Chemistry`, `Geography`, `Math`, and `Physics`) and Tech and Engineering (`Architecture & Engineering`, `Computer Science`, `Electronics`, `Energy & Power`, `Materials`, and `Mechanical Engineering`). This tailored subset comprises a total of 5,210 images.

**Implementation Details.** We evaluate the performance of the main-stream pre-trained LLMs using `GPT-4`, `GPT-4o`, and `GPT-4o-mini` as backbone models. Specifically, we utilize *gpt-4-turbo*, *chatgpt-4o-latest*, and *gpt-4o-mini-2024-07-18* respectively. The primary hyperparameters employed in our Chain-of-Region method include the pre-defined recognition call limits in split step and the cluster number during the unstructured merge step, detailed in Sections 3.1.2 and 3.1.3. In the current version, we let the base model used in the intermediate structure recognition call be the same as the one used in the final prediction. We set the recognition call limit to 10 and the cluster number to 5. We also provide the sensitivity analysis in this section. While other hyperparameters exist for structure split/merge steps, we adhere to either the default parameters provided by OpenCV or those commonly used in existing GitHub libraries. Given their minor impact on the final algorithm, we do not engage in an extensive discussion on these parameters.

### 4.1 MAIN RESULTS

**Baselines.** We evaluate diverse strategies for prompting VLMs for fine-grained analysis:

1. **Raw VLM model**: We sent the image and question directly into VLM to generate the final answer.

2. **Zero-shot CoT**[3] (Kojima et al., 2022) enhances the LLM's inference capabilities by including the phrase "*Let's think step by step*" in the prompt. We adapt this technique to the multimodal QA setting by modifying the prompt to "*Let's summarize information region by region before answering the question.*"

3. **Few-shot CoT** (Wei et al., 2022) leverages examples within the prompt to further improve LLM reasoning. In our approach, we incorporate examples of region descriptions in Appendix A.5.

---

[3]Alias for Chain-of-Thought.

4. **Meta Segment Anything Model 2 (SAM2)** (Research, 2024) offers a commercial-level segmentation tool to supply region masks. We employ this technique as a baseline for comparing segment results with our proposed CoR method. Specifically, we utilize the "everything" mode to provide initial segment pieces, applying the unstructured clustering/merging strategy outlined in Section 3.1.3 to produce the final regions, which are then processed in the information collection phase described in Section 3.2.

5. **SOM** (Yang et al., 2023) appends numbered tags to regions segmented by SAM2. The entire tagged image is then sent to the VLM for inference.

**CoT helps improve better performance in multimodal scientific question answering.** In Table 1, we present the primary results of our experiments, which compare the proposed Chain-of-Region (CoR) methods against baseline models (GPT4o-mini, GPT4-Turbo, and GPT-4o) across various tasks. Our observations are as follows: a) Zero-shot CoT and Few-shot CoT (Wei et al., 2022) enhance the VLM's inferential capabilities by eliciting a thought process that guides the VLM through problem-solving. b) SoM (Yang et al., 2023) proves to be less effective, primarily because it was originally designed for natural images, not scientific diagrams. The tags used in SoM can often mislead the VLM, leading to confusion and incorrect inferences. c) We also evaluated the baseline method SAM2 (Research, 2024), which employs the same inferential strategy to ours by sequentially summarizing regions, albeit using different segmentation masks. The results indicate that SAM2 is highly competitive with other baselines, underscoring the effectiveness of gathering information region-by-region prior to responding. However, CoR outperforms SAM2 by utilizing superior regional masks that better isolate and identify relevant features within diagrams. Overall, the CoR demonstrates its superior ability to handle complex visual information and contribute to more accurate VLM inferences.

**Hyperparameters and Sensitivity Analysis.** In our framework, two key hyperparameters are pivotal: a) the predefined recognition call limit in the split step as detailed in Section 3.1.2, and b) the number of clusters during the unstructured merge step, which is elaborated in Section 3.1.3. In the current implementation, we have set the recognition call limit to 10 and the number of clusters to 5. To evaluate sensitivity, we varied the recognition call limit within the range of $[3, 5, 10, 15, 20]$ and the cluster number within the range of $[1, 3, 5, 10, 20]$. We maintained other hyperparameters constant to isolate the effects of each tested variable.

The performance comparisons for each hyperparameter are illustrated in the bar plot in Figure 4. From this analysis, we observe that the recognition call limit has a relatively minor influence on the final outcomes. In contrast, the number of clusters significantly impacts performance as it controls the granularity of details to which the VLM attends. As shown in the bar plot, optimal performance is achieved at a moderate level of granularity; performance improves initially with an increase in cluster number but deteriorates when the number becomes excessively high. It aligns with the intuition: if the cluster number is too low, the VLM may overlook critical details; conversely, if it is too high, the VLM may become overwhelmed with minutiae and lose sight of the overall structure. Our results suggest that the ideal number of clusters should be tailored to the complexity of the input diagram. Dynamically selecting the cluster number opens an avenue for further research.

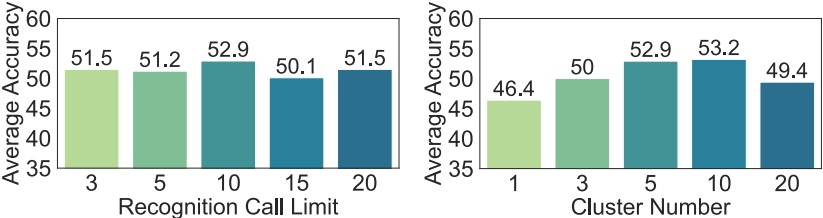

Figure 4: Sensitivity analysis of hyper-parameters on MMMU dataset. The average accuracy is reported on the MMMU dataset with backbone GPT-4o. The middle bar corresponds to the value used in our main experiments.

## 4.2 ANALYSIS ON SEGMENTATION RESULTS

In the realm of scientific diagram analysis, the precision in segregating each regional visual element is crucial. This section aims to compare the segmentation results of our method against Segment Anything Model 2 (SAM2) (Research, 2024), the state-of-the-art segmentation model but primarily

Table 1: **Main results for QA on scientific diagrams across the MMMU dataset.** `Bio.`, `Chem.`, `Geo.`, `Arch.`, `CS`, `Elec.`, `Mater.` and `ME` are short aliases for Biology, Chemistry, Geography, Architecture & Engineering, Computer Science, Electronics, Energy & Power, Materials, and Mechanical Engineering respectively.

| Method | Science | | | | | Tech and Engineering | | | | | | Average |
|---|---|---|---|---|---|---|---|---|---|---|---|---|
| | Bio. | Chem. | Geo. | Math | Physics | Arch. | CS | Elec. | Energy | Mater. | ME | |
| GPT4o-mini | 40.9 | 35.5 | 52.3 | 47.2 | 40.0 | 25.8 | 33.4 | 34.6 | 30.6 | 42.0 | 31.9 | 37.6 |
| + Zero-shot CoT | 38.4 | 32.6 | 52.1 | 38.7 | 45.7 | 29.2 | 35.1 | 23.1 | 34.7 | 42.4 | 44.4 | 37.9 |
| + Few-shot CoT | 31.3 | 37.7 | 50.9 | 45.1 | 42.3 | 30.4 | 45.5 | 31.0 | 47.5 | 38.7 | 35.0 | 39.6 |
| + SAM2 | 38.5 | 40.0 | 54.8 | 52.2 | 47.9 | 40.5 | 34.7 | 35.3 | 25.3 | 45.5 | 41.9 | 41.5 |
| + SoM | 31.2 | 33.6 | 40.0 | 43.5 | 45.1 | 35.2 | 27.0 | 17.9 | 41.2 | 34.2 | 29.9 | 34.4 |
| **+ CoR (Ours)** | 42.0 | 34.6 | 51.8 | 51.4 | 52.8 | 45.1 | 43.3 | 33.5 | 48.7 | 44.9 | 51.9 | **45.4** |
| GPT4-Turbo | 41.5 | 34.8 | 51.0 | 41.6 | 49.9 | 47.0 | 57.4 | 34.6 | 41.5 | 45.0 | 32.5 | 43.3 |
| + Zero-shot CoT | 43.3 | 25.1 | 59.6 | 46.5 | 58.3 | 44.8 | 50.8 | 31.3 | 46.8 | 31.4 | 50.0 | 44.3 |
| + Few-shot CoT | 40.0 | 37.8 | 51.8 | 52.8 | 52.4 | 51.6 | 57.9 | 47.6 | 45.2 | 32.2 | 51.7 | 47.4 |
| + SAM2 | 38.3 | 39.9 | 48.4 | 63.4 | 48.0 | 44.2 | 50.6 | 48.6 | 66.9 | 37.9 | 49.3 | 48.7 |
| + SoM | 41.0 | 30.3 | 44.8 | 47.6 | 44.0 | 42.3 | 37.5 | 15.1 | 42.1 | 26.9 | 24.7 | 36.0 |
| **+ CoR (Ours)** | 42.9 | 43.5 | 56.0 | 56.8 | 61.0 | 51.9 | 55.7 | 52.4 | 65.2 | 46.0 | 57.0 | **53.5** |
| GPT4o | 41.3 | 33.6 | 45.4 | 48.8 | 51.6 | 28.8 | 41.7 | 24.6 | 45.1 | 31.1 | 22.2 | 37.7 |
| + Zero-shot CoT | 47.7 | 46.1 | 47.4 | 52.2 | 66.6 | 44.3 | 50.6 | 29.3 | 41.5 | 42.6 | 47.5 | 46.9 |
| + Few-shot CoT | 47.8 | 38.6 | 58.0 | 44.9 | 56.1 | 51.3 | 35.2 | 34.2 | 41.8 | 36.5 | 42.3 | 44.2 |
| + SAM2 | 44.7 | 42.3 | 41.5 | 51.6 | 64.4 | 42.0 | 48.8 | 38.0 | 48.9 | 48.9 | 45.8 | 47.0 |
| + SoM | 48.9 | 37.3 | 44.7 | 55.1 | 49.8 | 25.9 | 38.9 | 28.1 | 57.5 | 33.9 | 24.6 | 40.4 |
| **+ CoR (Ours)** | 49.8 | 46.7 | 59.1 | 53.9 | 72.3 | 49.7 | 49.9 | 34.7 | 65.7 | 41.0 | 59.1 | **52.9** |

tailored for natural images, not scientific diagrams. SAM2 often overlooks small or slender objects, which are common in scientific diagrams. This section provides a comparison both quantitatively and qualitatively, highlighting the effectiveness of our approach in handling complex scientific diagrams.

**Groundth Truth Details.** Due to the lack of publicly available datasets on scientific diagram segmentation, we constructed a custom dataset derived from the MMMU. This dataset focuses specifically on instances where the raw predictions of GPT-4V (*gpt-4-turbo*) failed, thus representing challenging scenarios that test the robustness and precision of segmentation models. For our evaluation, we collected a total of 100+ samples, with an even distribution across all scientific categories. Further examples and visual illustrations of these segmentation approaches are provided in the Appendix B. We prepared two distinct types of segmentation masks for each image:

a) *Individual Segmentation Masks*: These masks are highly granular, with each individual visual element separated. For instance, the word "text" in a diagram is segmented into four distinct regions corresponding to each letter: 't', 'e', 'x', and 't'. This detailed segmentation allows us to precisely evaluate the accuracy of our split algorithm, gauging its ability to handle fine-grained details.

b) *Grouped Segmentation Masks*: In contrast, these grouped masks aggregate the visual elements into up to five cohesive semantic regions per image. This approach aims to assess how well the segmentation algorithm can identify and group related elements based on their semantic relationships.

**Metrics.** To evaluate the performance of our method CoR and SAM2, we used the *mean Intersection over Union* (mIoU) score as the primary metric, calculated for both the detailed and grouped segmentation masks. Unlike natural image segmentation tasks, our dataset lacks a one-to-one label mapping between ground truth and predicted regions. So, we use a region-matching strategy: for each ground truth region $M_{GT}$, we select the corresponding region $M$ in the prediction with the largest area of overlap. The IoU for this pair is calculated, and the process is repeated for all regions in the image. Once all individual IoU scores are computed, the *mIoU* for an image is determined by averaging the IoU values across all regions:

$$\text{mIoU} = \frac{1}{N} \sum_{i=1}^{N} |M_{GT_i} \cap M_i| / |M_{GT_i} \cup M_i|,$$

where $N$ is the total number of regions in the ground truth mask, $M_{GT_i}$ and $M_i$ represent the ground truth and predicted regions, respectively. For the $i$-th region, $|\cdot|$ denotes the area of the region.

**CoR Performs Better in Segmenting Scientific Diagrams than SAM2.** We measure the performance of both our segmentation method and SAM2 using the mIoU, calculated separately for the individual and grouped mask versions: *mIoU* (individual) quantifies the average overlap between the predicted masks and the ground truth in the *Individual Segmentation* setting, while *mIoU* (group) evaluates the average overlap for the *Grouped Segmentation* setting. The results are compiled in

Table 2: **Segmentation Accuracy on the MMMU Dataset.** It presents the mean Intersection over Union (mIoU) scores for both individual and grouped segmentation settings.

| Method | Science | | | | | Tech and Engineering | | | | | | Average |
|---|---|---|---|---|---|---|---|---|---|---|---|---|
| | Bio. | Chem. | Geo. | Math | Physics | Arch. | CS | Elec. | Energy | Mater. | ME | |
| mIOU (individual) | | | | | | | | | | | | |
| SAM2 | 36.8 | 47.7 | 30.1 | 39.9 | 56.4 | 39.9 | 44.7 | 52.5 | 30.4 | 22.7 | 37.4 | 42.2 |
| **CoR (Ours)** | 56.2 | 74.3 | 69.2 | 44.3 | 73.3 | 60.1 | 55.7 | 64.0 | 59.4 | 62.2 | 66.4 | 63.0 |
| mIOU (group) | | | | | | | | | | | | |
| SAM2 | 26.2 | 31.2 | 18.6 | 19.5 | 20.0 | 28.7 | 30.0 | 29.5 | 38.2 | 35.2 | 36.0 | 28.8 |
| **CoR (Ours)** | 40.0 | 45.4 | 40.4 | 34.1 | 39.0 | 36.3 | 41.3 | 43.5 | 46.7 | 31.9 | 57.2 | 41.8 |

Table 2, where we present both scores for both our method and SAM2 across the different categories of scientific diagrams. Notably, our method outperforms SAM2 in both metrics, with a significant margin (20.8% in the individual setting and 13.0% in the group setting). The numeric superiority highlights its fitness to scientific diagrams, making it a valuable tool for enhancing the capabilities of VLMs in scientific applications.

**Qualitative Comparison between SAM2 and CoR.** We supplemented our quantitative metrics with a qualitative visualization of the segmentation results in Figure 5 (more in Appendix B), focusing on direct visual comparisons between our method and SAM2. SAM2 struggled at providing accurate segmentation of fine lines and small annotations depicted in blocks like equations $s + 5/s + 7$, which are critical for fine-grained understanding. Our method, based on traditional computer vision techniques, is good at identified and segmented these intricate details with a clean boundary. These examples underscore the importance of precise segmentation in scientific analyses, particularly in capturing subtle yet crucial elements that contribute to accurate scientific diagram understanding.

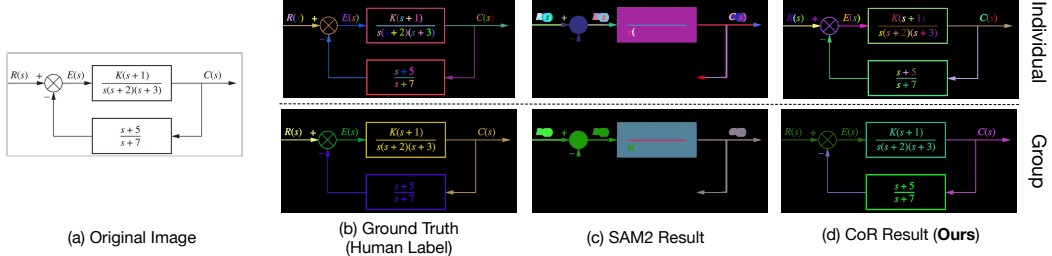

(a) Original Image    (b) Ground Truth (Human Label)    (c) SAM2 Result    (d) CoR Result (**Ours**)

Figure 5: Example of segmentation results on a scientific diagram. (a) Original image input. (b) Ground truth segmentations provided by human annotators. Two types of segmentations are shown: individual segmentation, which aims to capture all visual elements comprehensively, and group segmentation, which divides the image into up to five cohesive semantic regions. (c) Segmentation results generated by SAM2 (Research, 2024). (d) Results from the proposed Chain-of-Regions (CoR) approach. Each color in the masks represents the segment of a unique region. We provide more examples in Appendix B.

## 5 CONCLUSION

In this work, we have developed a comprehensive methodology to decompose regions and collect visual details in scientific diagrams using a combination of traditional computer vision techniques and VLM. Our approach, which involves initializing, splitting, and merging regions, shows significant improvements in the analysis and interpretation of complex scientific imagery.

**Limitations and Open Challenges.** While our approach has demonstrated promising results, there are limitations and open challenges worthy to share: (a) Our library of structured detectors, crucial for identifying and parameterizing visual elements, is not exhaustive. The path to obtaining hidden parameters of structured visual elements inherently involves reverse-engineering their creation processes, which poses a significant challenge. (b) Current DL methods also face limitations in this context. Unless they are specifically trained for a task, DL methods struggle to generalize across the varied structures encountered in scientific diagrams.

In a nut shell, our work represents a pioneering effort in combining traditional computer vision with modern machine learning techniques to tackle the problem of diagram analysis. Though the solutions we have developed are not universally perfect, we invite ongoing extensions to improve the applicability of our methods.

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

# A  METHODOLOGY DETAILS

## A.1  VLM-ASSISTED STRUCTURE RECOGNITION

In this section, we use GPT-4V to preliminarily identify potential structures or shapes within each main region. The input is prompted with the following instruction:

```
1 To better understand the details, you are shown only one part of the
      diagram at one time. Now, please summarize **all** the visual
      elements in 2-3 words.
2 - Please note that if there are multiple visual elements, use multiple
      lines and each line for one element.
3 - Each line starts with the color of the visual element.
4 - You should duplicate the line if there are multiple same visual
      elements.
```

An example of the input and output is shown in Figure 6. Detected keywords such as "arrow" and "circle" will trigger the corresponding structured splitting module, which we explain in more detail in Section A.2.

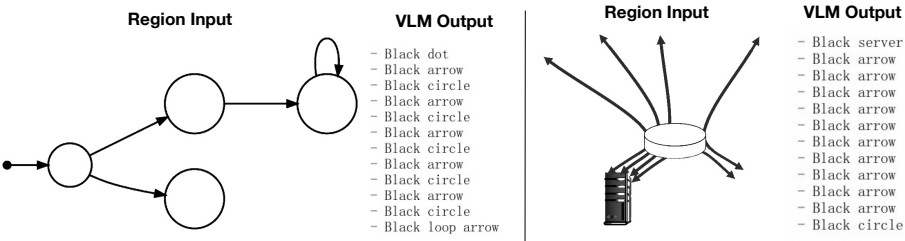

Figure 6: Two examples of region input and the corresponding output from GPT-4V.

## A.2  STRUCTURED SPLIT

We have incorporated a set of common structures typically encountered in the analysis of scientific diagrams. Below, we describe the detailed implementation of the `structure_detector` functions:

**Ellipses.** The `ellipse_detector` function is activated by the detection of keywords such as `circle`, `oval`, and `ellipse` during the Structure Recognition step (Appendix A.1). The implementation leverages the open-source package cited in (Meng et al., 2020). The meta information extracted includes parameters such as $(x_{center}, y_{center}, width, length, angle)$.

**Segments.** The `segment_detector` function is initiated by the detection of keywords `line`, `curve`, and `segment` during the Structure Recognition step (Appendix A.1). This functionality is facilitated by the open-source package referenced in (Von Gioi et al., 2008). The meta information extracted includes parameters such as $(x_{start}, y_{start}, x_{end}, y_{end}, width)$.

**Rectangles.** The `rectangle_detector` function is triggered by keywords `rect`, `square`, and `bar` identified in the Structure Recognition step (Appendix A.1). The implementation process involves: a) Detecting all segments using the package cited in (Von Gioi et al., 2008); b) Merging collinear segments; c) Detecting segment quadruplets that form a convex shape. The meta information extracted includes $(x_{center}, y_{center}, width, length, angle)$.

**Arrows.** The `arrow_detector` function is activated by the keyword `arrow`, as detected in the Structure Recognition step (Appendix A.1). Given that prior steps confirm the presence of an arrow shape and `segment_detector` determines the trajectory of the line or curve, the direction is then determined by calculating the position of `cv2.distanceTransform(M_i, cv2.DIST_L2).max()` and comparing the two endpoints to identify which is closer. The meta information extracted includes $(x_{start}, y_{start}, x_{end}, y_{end}, width)$.

The structured library can be further expanded by incorporating additional shape templates, enhancing the robustness and utility of our system for complex diagram analysis.

## A.3 STRUCTURED MERGE

In this section, we detail the implementation of structured region merging, leveraging both existing tools and heuristic rules.

**Captions.** We employ mature OCR tools (Contributors, 2024) to detect and generate caption boxes within diagrams. Regions enclosed by these text boxes are presumed to be related and are consequently merged into a unified region.

**Dotted lines.** Initially, we collect line segments previously detected by the `segment_detector` function, as introduced in Section A.2. Subsequently, we identify sequences of segments that meet the following criteria: a) The width and length of the boxes should be approximately equal (error <5%); b) The intervals between adjacent segment pairs should be similar (error <5%); c) The angular difference between the nearest pairs should be less than 20 degrees.

**Background lines.** Continuing from the same initial line segments detected by `segment_detector`, we focus on those with either horizontal or vertical orientations. For horizontal line segments, we compile a sequence of all y-axis positions, identifying the arithmetic mean with the largest gap and the longest contiguous segment. We then merge all horizontal lines whose y-axis values fall within the detected range. We apply an analogous process for vertical line segments, merging those within the same x-axis range.

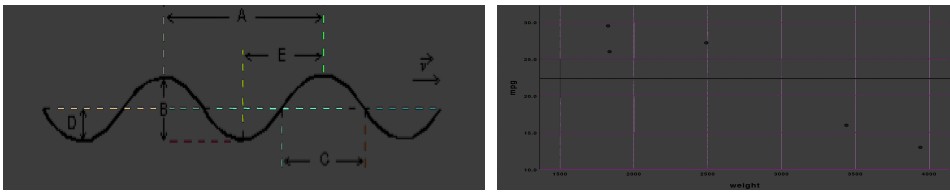

(a) Example of Dotted Line Merge                    (b) Example of Background Line Merge

Figure 7: Two examples of dotted line merging and background line merging respectively. The original image is faded, and the merged area is painted with different colors.

Figure 8: Illustration of how extracted information including structured information and semantic information is organized into an augmented prompt.

## A.4 SEMANTIC INFORMATION EXTRACTION

In this section, we extract semantic information that includes the role and entity type of each merged region with the following prompt:

```
1 To better understand the details, you are shown only one part of the
      diagram at one time. Now please **only** summarize the highlighted
      visual elements in short phrases.
```

### A.5 CHAIN-OF-THOUGHT PROMPT

In this section, we include the prompt used in the `Few-shot Chain-of-Thought` baseline:

```
1  [Original Question / Choices]
2
3   # Please summarize regions in the figure before answer the question.
       Here is the example of extracted region information:
4
5   Region [A]:  Circle labeled "P1";
6
7   Region [B]: Circular node labeled "P2";  Blue color for the node;  Solid
       black arrow pointing upwards to "P2"
8
9  Region [C]:  Circle labeled "P3" with one arrow pointing inward and one
      arrow pointing outward
10
11 Region [D]:  A node labeled "$P_4$" with one arrow pointing inward and
      one arrow pointing outward.
12
13 Region [E]:  Blue circle with a black border;  Text inside: "P5";  Arrow
      pointing towards the blue circle from below
```

## B  EXTRA VISUAL EXAMPLES

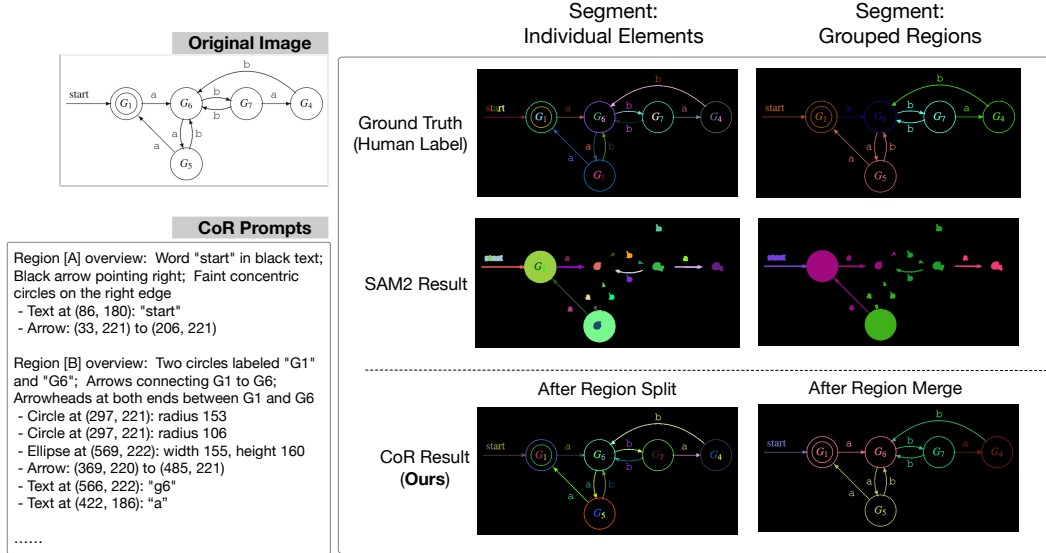

Figure 9: Example of CoR prompts and examples of segmentation results on a scientific diagram including: original image input, ground truth segmentations provided by human annotators, and segmentation results generated by SAM2 (Research, 2024) and the proposed Chain-of-Regions (CoR) approach.

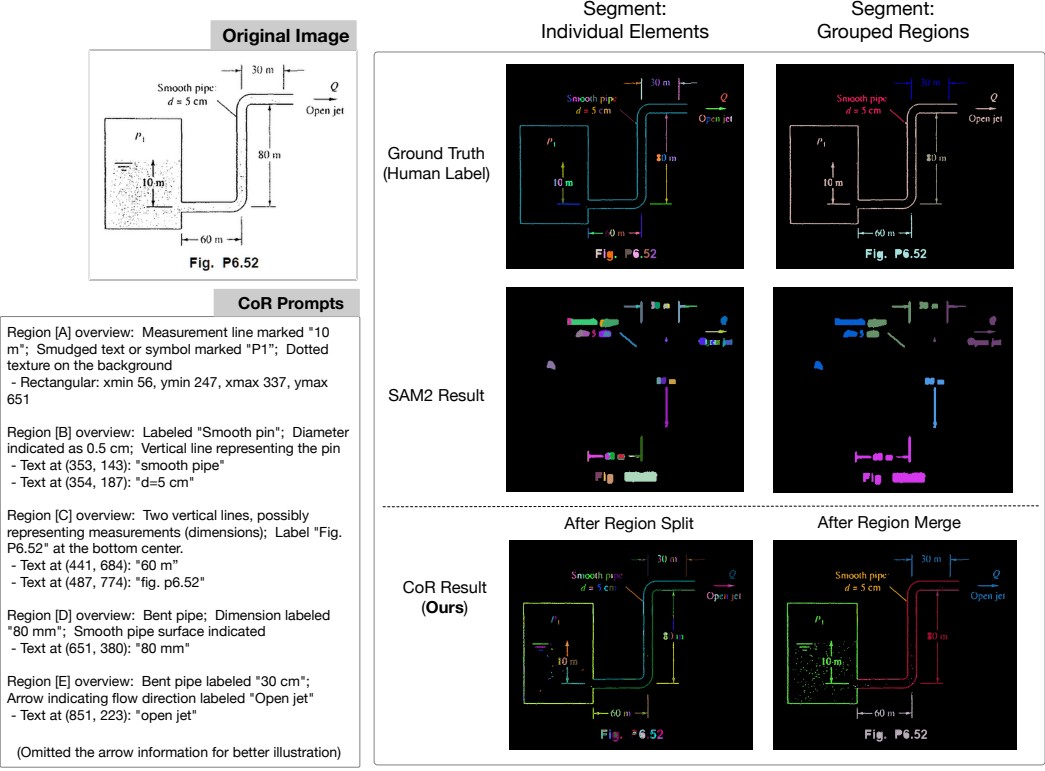

Figure 10: Example of CoR prompts and examples of segmentation results on a scientific diagram including: original image input, ground truth segmentations provided by human annotators, and segmentation results generated by SAM2 (Research, 2024) and the proposed Chain-of-Regions (CoR) approach.

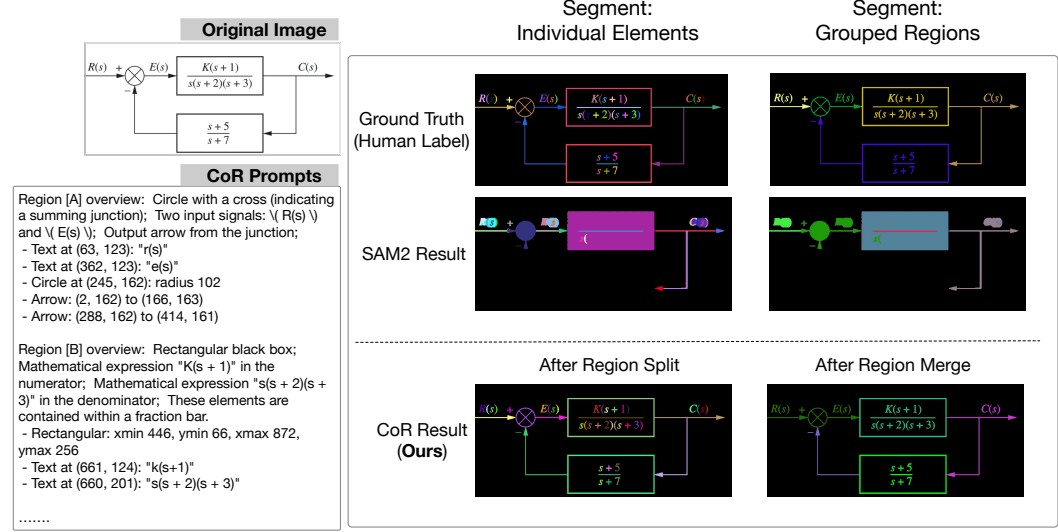

Figure 11: Example of CoR prompts and examples of segmentation results on a scientific diagram including: original image input, ground truth segmentations provided by human annotators, and segmentation results generated by SAM2 (Research, 2024) and the proposed Chain-of-Regions (CoR) approach.

