# OpenReview forum: "Chain-of-region: Visual Language Models Need  Details for Diagram Analysis"
_ICLR.cc/2025/Conference — ICLR 2025 Poster_

### Official Review · Reviewer_fCtq · 2024-10-29

**Soundness:** 3
**Presentation:** 3
**Contribution:** 2
**Rating:** 6
**Confidence:** 3

**Summary:**

This paper presents Chain-of-Region, a prompting method that enhances the capabilities of VLMs in scientific diagram analysis. This is achieved by cropping out regions of the diagram with traditional computer vision techniques based on heuristics and incorporating the extracted information into a master query. The effectiveness of this method is verified by evaluating the QA accuracy and the segmentation results on the MMMU dataset.

**Strengths:**

1. Practical methodology. This paper proposes a practical method for leveraging VLMs in scientific diagram analysis, allowing for plug-and-play generalization across models.
2. Significant performance gain. The method shows notable performance gains when applied to state-of-the-art models like GPT-4o.
3. Clear presentation. The paper is generally well-written, with clear figures that aid comprehension.

**Weaknesses:**

1. Limited generalizability. The method contains various heuristics tailored for scientific diagrams, limiting its applicability to more complex images. Extending this method for segmenting such images can be very challenging, if not impossible.
2. Unclear intuition. The rationale for how adding information from regions enhances QA task performance is not clear. While the example in Fig. 1 demonstrates how prompts lead the VLM to predict values based on the numbers in the prompts, I can’t see how they can aid the reasoning of various types of diagrams, such as the node-link diagram in Fig. 2. Providing more examples of the QA results and analyzing them can help better clarify this.
3. Unfair evaluation. The evaluation in Sec. 4.2 appears biased, as ground truth annotations favor the proposed method. The human labels only include foreground masks, penalizing methods that incorporate background masks, even when the cropped regions are identical (see the masks for K(s+1)/s(s+2)(s+3) as an example). A more appropriate evaluation method would use the area of the ground truth $|M_{GT_{i}}|$ as the denominator in the mIoU calculation.

**Questions:**

1. Why are background masks included in the region initialization? They appear to contain minimal information in the provided examples. Also, would they be regarded as the “main regions with large area size” in the VLM-assisted structure recognition?
2. How beneficial is the unstructured split and merge approach for analyzing scientific diagrams? Illustrations requiring this method are often considered single entities, as shown with the server in Fig. 6.

---

> ### Author Response · Authors · 2024-11-20
> **Response -- Part 1**
>
> We thank the reviewer for the insightful questions! Below we address each of your comments in detail.
>
> ---
>
> > ### Generalizability Beyond Scientific Diagrams
>
> **Original Comments**:
>
> *"Limited generalizability. The method contains various heuristics tailored for scientific diagrams, limiting its applicability to more complex images. Extending this method for segmenting such images can be very challenging, if not impossible."*
>
> **Response**:
>
> Fair concern! We acknowledge the limitation of relying on OpenCV-based algorithms for segmentation, particularly in their generalizability to more complex images beyond scientific diagrams. However, we see our approach as complementary to deep learning-based methods like SAM2, rather than as a standalone solution. Deep learning models, including SAM2, often struggle with scientific diagrams rich in intricate details such as arrows, intersecting lines, and subtle region boundaries. These features are critical for accurate interpretation but are challenging for methods trained primarily on natural image datasets. Our CoR framework leverages OpenCV's algorithms to address these unique challenges, enabling better handling of the structural nuances in scientific diagrams.
>
>
> > ### Rationale for how adding information from regions enhances QA task performance
>
> **Original Comments**:
>
> *"The rationale for how adding information from regions enhances QA task performance is not clear. While the example in Fig. 1 demonstrates how prompts lead the VLM to predict values based on the numbers in the prompts, I can’t see how they can aid the reasoning of various types of diagrams, such as the node-link diagram in Fig. 2. Providing more examples of the QA results and analyzing them can help better clarify this."*
>
> **Response**:
>
> Great questions! We acknowledge the need to clarify how adding region-based information improves QA performance, especially for complex diagrams like node-link diagrams in Fig. 2. Specifically, VLMs are prone to perceptual errors, which can mislead downstream QA tasks. For instance, in node-link diagrams, VLMs might confuse arrow directions or hallucinate nonexistent links. By providing pixel location data, these errors can be significantly reduced. Pixel locations allow VLMs to reason about the spatial relationships between nodes, ensuring that links or arrows are correctly interpreted. Similarly, in diagrams with captions—a common feature in scientific visuals—pixel locations of captions and their corresponding objects enable precise one-to-one mapping, reducing mismatches and enhancing reasoning.
>
> Beyond incorporating meta-information, our CoR framework inherently aligns with the divide-and-conquer paradigm. Even without additional metadata (e.g., as in SAM2 baseline in our paper), dividing diagrams into regions and processing them iteratively enables step-by-step inference. This approach mirrors chain-of-thought reasoning in text-based models, but adapted to the visual domain. By focusing on smaller regions sequentially, the CoR framework encourages VLMs to process complex diagrams in a step-by-step manner, resulting in substantial improvements in QA accuracy.

---

> ### Author Response · Authors · 2024-11-20
> **Response -- Part 2**
>
> > ### Discussion on the Evaluation in Section 4.2
>
> **Original Comments**:
>
> *"The evaluation in Sec. 4.2 appears biased, as ground truth annotations favor the proposed method. The human labels only include foreground masks, penalizing methods that incorporate background masks, even when the cropped regions are identical (see the masks for K(s+1)/s(s+2)(s+3) as an example). A more appropriate evaluation method would use the area of the ground truth as the denominator in the mIoU calculation."*
>
> **Response**:  Thank you for the feedback! The labeling was performed with a reasonable, commonsense judgment for each individual element, as exemplified in Figure 5(b). It is important to note that the performance gap between our method and SAM2 remains significant and is unlikely to be substantially affected by variations in labeling. This is because SAM2 struggles to discern finer details, particularly in scientific diagrams. For instance, as illustrated in Figure 5, SAM2 completely misses the line and a large block of the equation, which are critical components in this context.
>
> To further address the suggestion, we conducted an additional evaluation where the denominator in the mIoU calculation is replaced by the area of the ground truth. The updated results, shown in the table below, confirm that the CoR method remains more competitive, maintaining its advantage even with the revised metric.
>
>
> ---
>
> **Table: mIoU Comparison Across Categories with Updated Evaluation Method**
>
> | **Setting**   | **Method**   | **Bio.** | **Chem.** | **Geo.** | **Math** | **Physics** | **Arch.** | **CS**   | **Elec.** | **Energy** | **Mater.** | **ME**   | **Average** |
> |---------------|--------------|----------|-----------|----------|----------|-------------|-----------|----------|-----------|------------|-----------|----------|------------|
> | **Individual**| **SAM2**  | 63.15  | 75.92    | 46.44   | 67.43   | 73.19      | 68.10    | 79.14   | 76.07    | 68.79     | 45.61    | 75.03  | 67.17     |
> |               | **Ours**  | 67.91   | 89.69    | 83.67   | 57.09   | 88.26      | 72.66    | 67.97   | 81.07    | 72.15     | 75.70    | 80.20  | 76.03 |
> | **Group**     | **SAM2**     | 35.28   | 49.10    | 20.54   | 40.69   | 22.80 | 44.48    | 52.76   | 39.80    | 51.33     | 50.98    | 70.41  | 43.47     |
> |               | **Ours** | 47.05   | 56.40    | 59.03   | 51.59   | 60.82 | 58.81    | 54.42   | 61.28    | 62.06     | 52.97    | 67.33  | 57.43     |
>
> ---
>
> > ### Background Mask Used in Region Initialization
>
> **Original Comments**:
>
> *"Why are background masks included in the region initialization? They appear to contain minimal information in the provided examples. Also, would they be regarded as the “main regions with large area size” in the VLM-assisted structure recognition?"*
>
> **Response**:
>
> Great question! By isolating the foreground elements such as lines and captions, the remained image can contain distinct connected regions along with the background, like solid shapes with filled colors (different from the primary background color). To further clarify, only the main background (i.e., the largest outermost connected region) is excluded from the VLM-assisted structure recognition process, ensuring that our framework prioritizes significant visual components.
>
>
> > ### Effectiveness of unstructured split and merge approach for scientific diagram analysis
>
> **Original Comments**:
>
> *"How beneficial is the unstructured split and merge approach for analyzing scientific diagrams? Illustrations requiring this method are often considered single entities, as shown with the server in Fig. 6."*
>
> **Response**:
>
> Great question! To address the benefits and implications of the unstructured split and merge approach for analyzing scientific diagrams, we provide insights and empirical evidence below:
>
> * **Analysis for Merging**: Figure 4 illustrates our parameter analysis of the merging stage. The key intuition is achieving a balance: if the cluster number is too low, the VLM may overlook critical details; conversely, if it is too high, the VLM may become overwhelmed with minutiae and lose sight of the overall structure.
>
> * **Ablation Study on Unstructured Splitting**: To further evaluate the role of the unstructured splitting step, we conducted an ablation study where this step was removed. The Average performance with base model GPT4o is 50.1, indicating that excluding unstructured splitting leads to a performance drop (~3%) which highlights the importance of unstructured splitting.

---

> > ### Comment · Reviewer_fCtq · 2024-11-26
> >
> > I really appreciate the feedback and efforts from the authors. Some of the answers address my concerns, especially the additional experiments using the updated evaluation method. I am pleased to keep my current score (probably accept). Great work!

---

### Official Review · Reviewer_dTRw · 2024-11-02

**Soundness:** 3
**Presentation:** 3
**Contribution:** 3
**Rating:** 6
**Confidence:** 3

**Summary:**

This paper introduces the Chain-of-Regions (CoR) technique to enhance VLM's fine-grained recognition and reasoning capabilities for scientific charts. It demonstrates performance across disciplines including mathematics, architecture, and electronics. This paper combines established computer vision techniques with emerging topics in cross-modal understanding. I appreciate the approach of combining traditional computer vision methods to assist in enhancing the model’s performance, and indeed, the work shows a thorough consideration of various experimental details. The following  are some suggestions that may help the authors improve this work.
Traditional computer vision techniques, including the use of OpenCV, tend to be relatively sensitive to parameters. Yet, in the paper, it seems that authors have not analyzed the sensitivity of these parameters or how the understanding of diagram may vary across different scenarios.
Additionally, the rule-based reverse engineering appears to lack sufficient robustness—there has not been a comprehensive investigation to form the systematic theoretical framework for pixel recognition, segmentation, extraction, and integration. I suggest strengthening the discussion on this point.
Notably, sequential segmentation and rule-based combination are insufficient, as they may disrupt semantic context. I suggest that authors incorporate innovative algorithms in future work to further explore the interrelationships between different pixel regions.
In terms of writing, I suggest that the authors provide more pseudocode for the operations instead of directly presenting OpenCV code.
The parameter “Budget B” may also introduce uncertainty in the interpretation of subsequent diagram, but the authors have not provided much discussion on this.
In the Region Input of Appendix A1, we obtained the Output of different VLMs. It seems that this output is scanned from left to right. However, if the rotation angles of these images differ, what would the result be in that case?

**Strengths:**

This paper combines established computer vision techniques with emerging topics in cross-modal understanding. I appreciate the approach of combining traditional computer vision methods to assist in enhancing the model’s performance, and indeed, the work shows a thorough consideration of various experimental details.

**Weaknesses:**

The limitations of this paper lie in the potential impact of parameter sensitivity on the final interpretation, as well as whether the overall framework is adequately supported by methodological foundations.

**Questions:**

Traditional computer vision techniques, including the use of OpenCV, tend to be relatively sensitive to parameters. Yet, in the paper, it seems that authors have not analyzed the sensitivity of these parameters or how the understanding of diagram may vary across different scenarios.
Additionally, the rule-based reverse engineering appears to lack sufficient robustness—there has not been a comprehensive investigation to form the systematic theoretical framework for pixel recognition, segmentation, extraction, and integration. I suggest strengthening the discussion on this point.
Notably, sequential segmentation and rule-based combination are insufficient, as they may disrupt semantic context. I suggest that authors incorporate innovative algorithms in future work to further explore the interrelationships between different pixel regions.
In terms of writing, I suggest that the authors provide more pseudocode for the operations instead of directly presenting OpenCV code.
The parameter “Budget B” may also introduce uncertainty in the interpretation of subsequent diagram, but the authors have not provided much discussion on this.
In the Region Input of Appendix A1, we obtained the Output of different VLMs. It seems that this output is scanned from left to right. However, if the rotation angles of these images differ, what would the result be in that case?

---

> ### Author Response · Authors · 2024-11-20
> **Response**
>
> We thank the reviewer for the insightful questions! Below we address each of your comments in detail.
>
> ---
>
> > ### Support of theoretical foundations
>
> **Original Comments**:
>
> *"The rule-based reverse engineering appears to lack sufficient robustness—there has not been a comprehensive investigation to form the systematic theoretical framework for pixel recognition, segmentation, extraction, and integration. I suggest strengthening the discussion on this point."*
>
> **Response**:
>
> We are happy to provide more contexts here! Our approach indeed draws inspiration from the formal principles of a **Divide-and-Conquer** algorithm, which can be understood as follows:
>
> * *Region Initialization and Splitting*: These stages function similarly to the "Divide" phase, where a complex problem is broken into manageable sub-problems that can be analyzed or conquered independently.
> * *Region Merging*: This aligns with the "Combine" step, where the processed sub-regions are reassembled, ensuring that coherent segments of the image are combined to produce meaningful semantic information.
>
> From a cognitive science perspective, our method also parallels the `System 1 and System 2` theory. When VLMs process an image in a single pass, this resembles `System 1` thinking—a fast, automatic, intuitive mode. However, for complex scientific diagrams requiring detailed understanding of each component, `System 2` thinking mode ---- a slower, deliberate, and analytical approach, is more effective. Our CoR framework equips VLMs to approximate `System 2` thinking by enabling an iterative process of breaking down the thought steps into more careful consideration and then reassembling components, thereby enhancing comprehension of intricate visual information.
>
>
> > ### Parameter sensitivity analysis including budget parameter
>
> **Original Comments**:
>
> *"Traditional computer vision techniques, including the use of OpenCV, tend to be relatively sensitive to parameters. Yet, in the paper, it seems that authors have not analyzed the sensitivity of these parameters or how the understanding of diagram may vary across different scenarios."*
>
> **Response**:
>
> Thank you for raising this point! We address parameter sensitivity for CoR in Section 4.1, specifically in Figure 4. It includes the analysis for number of recognition call and the cluster number (also budge B). Regarding the OpenCV algorithms used in our framework, we acknowledge the reviewer's concerns about potential parameter sensitivity. To mitigate such issues, we employ approaches that minimize the need for manual parameter tuning. For example, we utilize adaptive thresholds like cv2.THRESH_OTSU, which dynamically adjust thresholds based on the image's characteristics, ensuring robustness across a variety of scenarios. Additionally, for other parameters, we rely on well-established default values that are broadly effective for most scientific diagrams. These parameters were not fine-tuned for individual images or datasets but were chosen by the default value to provide generalizable performance.
>
>
>
> > ### Discussion about the pixel region's Interrelationships
>
> **Original Comments**:
>
> *"I suggest that authors incorporate innovative algorithms in future work to further explore the interrelationships between different pixel regions. In terms of writing, I suggest that the authors provide more pseudocode for the operations instead of directly presenting OpenCV code."*
>
> **Response**:
>
> We appreciate the reviewer highlighting this open challenge of associating pixel regions in a semantically aware manner. In our current work, we adopt a straightforward strategy: we place all relevant information of regions into the VLM's context, leveraging the model's inherent reasoning capabilities to establish semantic relationships between regions. A promising direction for future work is to represent pixel-level information as token embeddings during the VLM's training phase. This could allow for a more integrated pipeline of combining segmentation information and the semantic structure. Finally, we are grateful for the suggestion to refine our writing and will revise the pseudocode to improve clarity of our work as suggested.
>
> > ### Question regarding scanning order for examples in Appendix A1
>
> **Original Comments**:
>
> *"In the Region Input of Appendix A1, we obtained the Output of different VLMs. It seems that this output is scanned from left to right. However, if the rotation angles of these images differ, what would the result be in that case? "*
>
> **Response**:
>
> In Figure 6, the input images are directly fed into the VLMs, and due to the black-box nature of models like GPT-4, it is challenging to definitively determine whether they employ an inherent scanning order mechanism, such as left-to-right processing. However, to address the reviewer’s query regarding rotation, we observed that the recognition results (like circles and arrows) remain consistent even when the images are rotated.

---

### Official Review · Reviewer_WY7p · 2024-11-03

**Soundness:** 3
**Presentation:** 3
**Contribution:** 3
**Rating:** 6
**Confidence:** 2

**Summary:**

This paper presents a method for scientific diagram understanding that combines traditional CV methods (i.e., region initialization, splitting, and merging) with VLM prompting. The approach is evaluated on two both MMMU scientific diagrams and a small-scale diagram segmentation dataset. Results indicate the superiority of the proposed method.

**Strengths:**

- I appreciate how this work recognizes that a key challenge of diagram understanding is parsing the details and proposes a method combining classical CV methods to complement VLMs.
- The authors benchmarked a wide array of prompting strategies in their first experiment.
- The overall narrative is easy to follow.

**Weaknesses:**

I want to preface this by saying that I am a relatively junior reviewer, so please take my comments with a grain of salt.
- The authors mentioned they developed some structured detectors to identify and parametrize visual elements in diagrams. The performance of the method also seems to depend on the diagram having homogeneous colors and structured patterns. This raises questions about the generalizability of the approach. In experiment 1, the authors only evaluated the method on MMMU alone. Therefore, I wonder how this partially heuristics-based method would fare on other datasets. Some extra experiments might be helpful for demonstrating the power of the method.
- Some crucial details about Experiment 2 (on segmentation) seem to be missing. How many diagrams are there exactly in the dataset used in experiment 2? The authors wrote "100+", but this seems very unrigorous... In addition, how were these diagrams selected? I noticed all examples shown for exp 2 are in grayscale, which leads me to wonder if they were specifically chosen to play to the strengths of the proposed method that is largely heuristics-based. In addition, only SAM2 was compared against CoR. Perhaps it would be helpful to include some additional baselines?

**Questions:**

1. How many diagrams are there exactly in the dataset used in experiment 2? The authors wrote "100+", but this seems very unrigorous... In addition, how were these diagrams selected? I noticed all examples shown for exp 2 are in grayscale, which leads me to wonder if they were specifically chosen to play to the strengths of the proposed method that is largely heuristics-based.
2. Can the authors provide a decomposition of the number of charts for each category in experiment 1 for the sake of clarity?
3. What is a rough cost comparison of the different prompting approaches in Experiment 1? If CoR is very expensive compared to other approaches, would some form of inference-time scaling (e.g., generating multiple responses and using majority voting) improve baseline method performance? I'm not sure if this is a valid thing to do, so please feel free to argue if this is a bad idea.

---

> ### Author Response · Authors · 2024-11-20
> **Response -- Method**
>
> We thank the reviewer for the insightful questions! Below we address each of your comments in detail.
>
> ---
>
> > ### Generalizability of the approach
>
> **Original Comments**:
>
> *"The authors mentioned they developed some structured detectors to identify and parametrize visual elements in diagrams. The performance of the method also seems to depend on the diagram having homogeneous colors and structured patterns. This raises questions about the generalizability of the approach."*
>
> **Response**:
>
> Our approach is tailored specifically for scientific figure understanding, where structured patterns are the norm. Most scientific diagrams inherently follow structured layouts, making our method broadly applicable in this domain. However, we acknowledge the limitations of the current version when it comes to diagrams that combine natural imagery with geometric elements, such as topographic maps or 3D model illustrations.
>
> Notably, our structure split library allows us to integrate new structure algorithms into our library to handle more visual elements while maintaining compatibility with existing components. Our research open avenues for future research and could broaden the applicability of our method beyond existing structured algorithms.
>
>
> > ### Experiments on other datasets than MMMU
>
> **Original Comments**:
>
> *"In experiment 1, the authors only evaluated the method on MMMU alone. Therefore, I wonder how this partially heuristics-based method would fare on other datasets. Some extra experiments might be helpful for demonstrating the power of the method."*
>
> **Response**:
>
> Great suggestion!
> While our primary focus was on the MMMU dataset due to its relevance to our research, we recognize the existence of other chart datasets like ChartQA. ChartQA is often used to benchmark methods trained specifically for chart-like data (e.g., ChartLlama), which covers a small subset of the scientific diagram. Although our method is not tailored exclusively for charts, CoR achieves performance comparable to state-of-the-art methods as shown in the following table.
>
>
> **Table: Accuracy Comparison on ChartQA**
>
> | Model    | Human Authored QA    | Machine Generated QA | Average  |
> | -------- | -------- | --------  | -------- |
> | Pix2struct [1] |  30.50 | 81.60 | 56.00 |
> | Matcha [2] | 38.20 | 90.20 | 64.20 |
> | Unichart [3] | 43.92 | 88.56 | 66.24 |
> | ChartLlama [4] | 48.96 | 90.36 | 69.66 |
> | GPT4o    | 60.71    | 78.04     | 69.38    |
> | + CoR      | 67.85    | 87.81     | 77.83    |
>
> **Reference**
>
> [1] Lee, K., Joshi, M., Turc, I. R., Hu, H., Liu, F., Eisenschlos, J. M., ... & Toutanova, K. (2023, July). Pix2struct: Screenshot parsing as pretraining for visual language understanding. In International Conference on Machine Learning (pp. 18893-18912). PMLR.
>
> [2] Liu, F., Piccinno, F., Krichene, S., Pang, C., Lee, K., Joshi, M., ... & Eisenschlos, J. M. (2022). Matcha: Enhancing visual language pretraining with math reasoning and chart derendering. arXiv preprint arXiv:2212.09662.
>
> [3] Masry, A., Kavehzadeh, P., Do, X. L., Hoque, E., & Joty, S. (2023). Unichart: A universal vision-language pretrained model for chart comprehension and reasoning. arXiv preprint arXiv:2305.14761.
>
> [4] Han, Y., Zhang, C., Chen, X., Yang, X., Wang, Z., Yu, G., ... & Zhang, H. (2023). Chartllama: A multimodal llm for chart understanding and generation. arXiv preprint arXiv:2311.16483.

---

> ### Author Response · Authors · 2024-11-20
> **Response -- Experiments**
>
> > ### Dataset Details on the Size of Sub-category
>
> **Original Comments**:
>
> *"Can the authors provide a decomposition of the number of charts for each category in experiement 1 for the sake of clarity?"*
>
> **Response**:
>
> Thank you for pointing this out! Our dataset split directly follows the sub-category divisions provided in MMMU. For the reviewer’s convenience, we provide the detailed number of scientific diagrams for each sub-category below:
>
>
>
> **Table: Number of Images for Subcategories**
>
> | **Category**          | **Bio.** | **Chem.** | **Geo.** | **Math** | **Physics** | **Arch.** | **CS** | **Elec.** | **Energy** | **Materials** | **ME** |
> |------------------------|-------------|---------------|---------------|----------|-------------|-------------------------------|-----------------------|-----------------|--------------------|---------------|---------------------------|
> | **No. of Images** | 345 | 603 | 565 | 505 | 408 | 551 | 371 | 256 | 432 | 458 | 429 |
>
>
> > ### Discussion on the Cost and Comparison with Baselines of Similar Cost
>
> **Original Comments**:
>
> *"What is a rough cost comparison of the different prompting approaches in Experiment 1? If CoR is very expensive compared to other approaches, would some form of inference-time scaling (e.g., generating multiple responses and using majority voting) improve baseline method performance? I'm not sure if this is a valid thing to do, so please feel free to argue if this is a bad idea."*
>
> **Response**:
>
> Great question! Here's a breakdown of the cost comparison and an analysis of your suggestion regarding inference-time scaling:
>
> * **Cost Computation**: The overall cost for our CoR framework can be expressed as:
> `Image Processing Time (CPU) + Number of API Calls × API Call Time.`
> Empirically, the image processing time by OpenCV is negligible compared to the time spent on API calls. Therefore, for fair comparison, we focus on methods that involve a similar number of API calls. Specifically, the "+SAM2" and "+SoM" baselines in our paper are designed to maintain the same clustering number (budget B) as CoR. We also include the numers in the table below for reviewer's convenience.
>
> * **Inference-Time Scaling (e.g., Majority Voting)**: As suggested, we also evaluated a baseline that uses majority voting at inference time. While majority voting can improve baseline performance in some cases, it does not significantly improve the overall performance. We include these results below for the reviewer’s reference.
>
>
> ---
>
> **Table: Model Performance Comparison between Majority Voting (n=5) and CoR**
>
> | Model                       | Bio. | Chem. | Geo. | Math | Physics | Arch. | CS   | Elec. | Energy | Mater. | ME   | Average |
> |-----------------------------|------|-------|------|------|---------|-------|------|-------|--------|--------|------|---------|
> | GPT4o-mini                  | 40.9 | 35.5  | 52.3 | 47.2 | 40.0    | 25.8  | 33.4 | 34.6  | 30.6   | 42.0   | 31.9 | 37.6    |
> | + Vote | 44.2 | 38.0 | 56.0 | 50.7 | 42.6 | 31.9 | 36.1 | 34.8 | 34.5 | 47.1 | 35.9 | 41.1 |
> | + CoR          | 42.0 | 34.6  | 51.8 | 51.4 | 52.8    | 45.1  | 43.3 | 33.5  | 48.7   | 44.9   | 51.9 | 45.4    |
> | GPT4-Turbo                  | 41.5 | 34.8  | 51.0 | 41.6 | 49.9    | 47.0  | 57.4 | 34.6  | 41.5   | 45.0   | 32.5 | 43.3    |
> | + Vote | 43.1 | 40.1 | 53.8 | 45.6 | 50.6 | 49.6 | 56.6 | 36.7 | 42.5 | 45.9 | 42.5 | 46.1 |
> | + CoR          | 42.9 | 43.5  | 56.0 | 56.8 | 61.0    | 51.9  | 55.7 | 52.4  | 65.2   | 46.0   | 57.0 | 53.5    |
> | GPT4o                       | 41.3 | 33.6  | 45.4 | 48.8 | 51.6    | 28.8  | 41.7 | 24.6  | 45.1   | 31.1   | 22.2 | 37.7    |
> | + Vote | 47.5 | 36.7 | 55.8 | 53.6 | 59.6 | 46.1 | 56.4 | 27.4 | 45.9 | 37.5 | 26.6 | 44.8 |
> | + CoR               | 49.8 | 46.7  | 59.1 | 53.9 | 72.3    | 49.7  | 49.9 | 34.7  | 65.7   | 41.0   | 59.1 | 52.9    |
>
>
> ---
>
>
> * As a side note, regarding API cost, CoR does not introduce a substantial overhead compared to raw VLM usage. Most VLMs, such as GPT-4 vision variants, calculate cost based on the number of pixels processed. While CoR processes images region by region, the cumulative pixel count remains nearly equivalent to sending the entire image in a single batch. Empirically, we observe that CoR’s API cost is approximately 3× that of the raw model, a reasonable trade-off given the improvement in performance.

---

> ### Author Response · Authors · 2024-11-20
> **Response -- Discussion**
>
> > ### Detailed Discussion around Segmentation Experiments
>
> **Original Comments**:
>
> *"How many diagrams are there exactly in the dataset used in experiment 2? The authors wrote "100+", but this seems very unrigorous… In addition, how were these diagrams selected? I noticed all examples shown for exp 2 are in grayscale, which leads me to wonder if they were specifically chosen to play to the strengths of the proposed method that is largely heuristics-based. In addition, only SAM2 was compared against CoR. Perhaps it would be helpful to include some additional baselines?"*
>
> **Response**:
>
> * **Dataset Size of Segmentation Comparison**: The segmentation dataset used in Experiment 2 comprises 154 diagrams. While this number is currently modest due to the high cost of detailed mask labeling, we recognize the importance of a larger dataset and plan to expand it in the future. We also encourage the research community to develop and release a public segmentation dataset focused on scientific diagrams, which would greatly benefit this area of study. Despite the limited dataset size, our experiments consistently demonstrate the advantages of CoR over SAM2, highlighting the robustness of our framework.
>
> * **Image Selection and the Grayscale Prevalence**: The selection of diagrams was purely formed by the failure cases of GPT-4 to collect challenging examples, rather than any preference that would artificially boost the performance of our method. The prevalence of grayscale images in our dataset reflects the common nature of scientific diagrams, such as circuit diagrams and math questions from textbooks, which are often grayscale. This was not a deliberate choice to favor CoR.
>
>     Furthermore, our method does not exhibit a preference for grayscale over colored images, as evidenced by results on colored diagrams shown in Figures 2 and 8 of the paper. SAM2's struggles with scientific diagrams are pervasive, regardless of grayscale or color. To corroborate this, we also welcome reviewers to test random diagrams like in papers on the public portals from META. While SAM2 excels at segmenting natural images, our results demonstrate the unique advantages of combining OpenCV and CoR for scientific diagrams.
>
> * **Baselines and Comparisons**: SAM2 was chosen as a baseline because it represents a well-established state-of-the-art segmentation method and has demonstrated significant improvements over its predecessor, SAM1, since its release in August 2024. Other general segmentation models are way worse than it.
>
>     To the best of our knowledge, no publicly available segmentation models have been specifically trained on scientific diagrams. However, we are open to any specific baseline suggestions that the reviewer may propose. The lack of competitive public segmentation models for this niche domain further emphasizes the need for tailored solutions like CoR.

---

### Official Review · Reviewer_sLjr · 2024-11-04

**Soundness:** 3
**Presentation:** 3
**Contribution:** 3
**Rating:** 6
**Confidence:** 4

**Summary:**

This paper developed a new method called "Chain-of-Regions." This method is to help language models to accurately understand the visual information present in a figure such as scientific graphs or diagrams. While general models like GPT have great capabilities to understand images accurately, some gaps need to be filled. The novelty of this paper lies in that the authors adopted traditional CV techniques (not deep learning models) to decompose the image before feeding the broken-down parts into the visual language model.

The proposed CoR method involves a three-step process. The first step (Region Initialization) is to break down regions using connected components analysis offered by OpenCV. This initially identified regions further analyzed in the second step (Region Splitting). The second step further identifies detailed components in each region by checking whether common geometric shapes exist or unstructured shapes that can be identified from the liquid filling analogy. Lastly, the third step (Region Merging) finally checks how many regions can be practically processed in order to make it computationally efficient enough.

Based on the proposed CoR method, the authors report significant improvement in performance in region segmentation. The authors also conducted various ablation studies that confirms each step's contribution to the overall performance enhancement. With regard to implementation, the refreshing point is that this CoR method can be run on CPU only.

**Strengths:**

1) The novel and refreshing contribution of this paper is that employing old computer vision techniques in a smart way can be very helpful with modern visual language models. Inspired by this paper, future studies may look into such possibilities to replace certain parts of the language model pipeline.

2) The nice thing about the CoR method is that it can be solely run on CPUs (because it only uses OpenCV's connected component analysis and shape detection algorithm), which means it's very cost-effective and probably fast too.

3) The results look solid, as shown by the extensive ablation studies.

4) Another nice property of this method is that it can be used in a Plug-and-Play fashion with the existing visual language model pipeline. This is actually a good property because it can actually have some practical impact in this ever-growing LLM era. The proposed method's versatility in identifying different shapes also makes it easily adopted in the operating pipeline as a new component.

5) Lastly, this proposed method is clearly a white-box approach, which is transparent in showing why the regions were split in a certain way.

**Weaknesses:**

1) One thing that I find a bit short is the theoretical justification of why the authors made certain technical and/or methodological choices. I think the implementation and the practical implications are strong, but the authors fall short of explaining why their approach works better. The three steps of Region Initialization, Region Splitting, and Region Merging make sense to me, but having a more well-written formal justification of these steps would make the paper rock solid.

2) While the authors compare the CoR method with SAM2, I wonder how the CoR method performs compared to other specialized models for processing scientific diagrams. While impressive, the results would be more robust if the authors could present a bit more head-to-head comparisons with competing models and/or approaches.

3) The results are unanimously great, which is good of course, but it also makes me wonder what is the boundary condition and where this approach fails. Currently, the results look just too good. Knowing when and where a model fails is very helpful for future studies to build upon. In addition, the limitations that the authors acknowledge at the end are just nominal---there may be some other geometric shapes in our library. I want to hear about the real limits of this approach.

4) Compared to Steps 1 and 2, I think Step 3 (Region Merging) needs a bit more clarification. It involves budget parameter B, but I don't find much justification or rigorous discussion on the choice of B. Discussing at least the trade-off between a large and a small B would be nice.

5) Since this paper's core novelty lies in the combination of a traditional method with a modern VLM, the authors need to explain better how the output of OpenCV and CoR is fed into and/or interacted with the VLM.

**Questions:**

1) How does the proposed method fare on parsing out real-world scientific diagrams?

2) Did you consider using other operators or functionalities of OpenCV as alternatives to the connected components analysis and the shape detection algorithm?

3) The structured splitting part of Step 2 involves identifying commonly used geometric shapes. What shapes are included in the library, and how comprehensive are they in your assessment?

4) All the performance reported in the paper is about accuracy (correct me if I am wrong). However, the authors push cost and computation efficiency (CPU only) as the main strengths of their approach. Do you have any hint of comparisons about those computational complexities and how they scale?

5) As I mentioned in the weaknesses, I am particularly interested in learning more about the interface between OpenCV and VLM. How did you ensure the regions detected by OpenCV are correctly interpreted by the VLM downstream? Were there any possibilities of conflict between the two modules?

---

> ### Author Response · Authors · 2024-11-20
> **Response -- Method Part 1**
>
> We thank the reviewer for the insightful questions! Below we address each of your comments in detail.
>
> ---
>
> > ### Formal Intuition of CoR Method
>
> **Original Comments**:
>
> "*One thing that I find a bit short is the theoretical justification of why the authors made certain technical and/or methodological choices. I think the implementation and the practical implications are strong, but the authors fall short of explaining why their approach works better. The three steps of Region Initialization, Region Splitting, and Region Merging make sense to me, but having a more well-written formal justification of these steps would make the paper rock solid.*"
>
> **Response**:
>
> Thank you for the insightful suggestions! Our approach indeed draws inspiration from the formal principles of a **Divide-and-Conquer** algorithm, which can be understood as follows:
>
> * *Region Initialization and Splitting*: These stages function similarly to the "Divide" phase, where a complex problem is broken into manageable sub-problems that can be analyzed or conquered independently.
> * *Region Merging*: This aligns with the "Combine" step in Divide-and-Conquer, where the processed sub-regions are reassembled, ensuring that coherent segments of the image are combined to produce meaningful semantic information.
>
> In the context of transformers, the Chain-of-Region (CoR) framework reflects the power of Chain-of-Thought (CoT) reasoning, particularly valuable for transformers with limited depth. CoT allows transformers to decompose complex tasks into modular steps, which in theory provides a stronger expressive upper bound. Specifically, as detailed in [1], constant-depth transformers with constant-bit precision, can only supports tasks within AC0 without decomposition. However, using CoT for T steps enables constant-depth transformers to solve any problem solvable by boolean circuits of size T.
>
> From a cognitive science perspective, our method also parallels the `System 1 and System 2` theory [2]. When VLMs process an image in a single pass, this resembles `System 1` thinking—a fast, automatic, intuitive mode. However, for complex scientific diagrams requiring detailed understanding of each component, `System 2` thinking mode ---- a slower, deliberate, and analytical approach, is more effective. Our CoR framework equips VLMs to approximate `System 2` thinking by enabling an iterative process of breaking down the thought steps into more careful consideration and then reassembling components, thereby enhancing comprehension of intricate visual information.
>
> References:
>
> [1] Li, Z., Liu, H., Zhou, D., & Ma, T. (2024). Chain of Thought Empowers Transformers to Solve Inherently Serial Problems. The Twelfth International Conference on Learning Representations.
>
> [2] Kahneman, D. (2011). Thinking, Fast and Slow. Farrar, Straus and Giroux.
>
>
> > ### Interaction Detail between OpenCV and VLM
>
> **Original comment**:
>
> "*How did you ensure the regions detected by OpenCV are correctly interpreted by the VLM downstream? Were there any possibilities of conflict between the two modules?*"
>
> **Response**:
>
> Great points! We’ve structured our response as follows:
>
> * **Incorporating OpenCV Outputs into VLM**:
> The integration of OpenCV results to VLM is achieved by directly incorporating structured semantic information, such as pixel locations and component boundaries, into the input context. For example, as shown in Figure 8, details like connectivity between nodes are made explicitly accessible in the context, enabling the VLM to leverage this spatial information for more precise interpretation and accurate answers.
>
> * **Interpretation of OpenCV-Detected Regions by the VLM**:
> OpenCV's role is to supplement the VLM with specific, contextual details crucial for certain tasks. Our empirical results demonstrate that VLMs effectively utilize this additional information, resulting in improved accuracy. Recent studies indicate that large language models, when presented with explicit contextual evidence, tend to prioritize this information over their own priors, which further supports our approach [3].
>
> * **Handling Potential Conflicts Between OpenCV and VLM**:
> In the CoR framework, OpenCV and the VLM work collaboratively, particularly in the Region Splitting phase (Section 3.1.2). Here, the VLM first identifies shapes (e.g., circles) within a region, and this preliminary recognition is then passed to OpenCV's structure-splitting algorithms (e.g., ellipse detection). If the VLM's recognition is incorrect, then the structured split algorithm will fail to find the shape, and its metadata will not be included in the context. Empirically, the likelihood of such conflicts is low, as the VLM at this stage mainly processes small and non-complex sub-regions.
>
> References:
>
> [3] Shi, Z., Wei, J., Xu, Z., & Liang, Y. Why Larger Language Models Do In-context Learning Differently?. In R0-FoMo: Robustness of Few-shot and Zero-shot Learning in Large Foundation Models.

---

> ### Author Response · Authors · 2024-11-20
> **Response -- Method Part 2**
>
> > ### Alternatives to the connected components analysis and the shape detection algorithm.
>
> **Original Comments**:
>
> *"Did you consider using other operators or functionalities of OpenCV as alternatives to the connected components analysis and the shape detection algorithm?"*
>
> **Response**:
>
> The connected component analysis identifies and labels contiguous groups of pixels with the same value, typically foreground pixels, in a deterministic manner for a given binary input image. Due to this inherent determinism, we did not actively seek alternative functions for this specific purpose. However, variations can be introduced by adjusting the connectivity definition:
> * *4-connectivity*: Considers only neighboring pixels in the up, down, left, and right directions.
> * *8-connectivity*: Includes all neighboring pixels, including diagonals.
> In our implementation, we utilized the default 8-connectivity setting. We tested both configurations and found no notable performance differences between the two.
>
> Regarding shape detection, we incorporated state-of-the-art and open-sourced algorithms in our structured split library. For rectangular and segment detection, we utilized the approach proposed by Von Gioi et al. (2008). For ellipse detection, we adopted the method developed by Meng et al. (2020). They provided reliable shape identification suited to our analysis needs.
>
> > ### Shape supported in the structure split library
>
>
> **Original Comments**:
>
> *"The structured splitting part of Step 2 involves identifying commonly used geometric shapes. What shapes are included in the library, and how comprehensive are they in your assessment?"*
>
> **Response**:
>
> We provide a detailed overview of the shape-detection algorithms in Appendix A.2, where we include commonly encountered shapes in scientific diagrams, such as ellipses, arrows, segments and rectangles. To broaden our approach, we’ve made efforts to extend the library with additional shapes like triangles and polygons. However, the capabilities of available public tools, such as OpenCV’s `cv2.approxPolyDP`, have inherent limitations—they work effectively only when shapes are already isolated. The constraint highlights an opportunity for further development in the field, and we hope our work draws attention to the continued value of traditional vision strategies.
>
> For shapes that are unstructured, like the hand and handle in Figure 3, we implemented a custom algorithm inspired by high-viscosity fluid filling to help separate complex regions. This approach helps to address instances where default methods fall short, ensuring greater robustness in handling irregular shapes.

---

> ### Author Response · Authors · 2024-11-20
> **Response -- Experiment & Discussion**
>
> > ### More Baseline Comparison
>
> **Original Comments**:
>
> *"The results would be more robust if the authors could present comparisons with competing models."*
>
> **Response**:
>
> Great suggestion! We compared our method with the baseline ChartLlama [4], which is specifically trained on chart-type diagrams. As the following table shows, the results indicate that the average accuracy of ChartLlama is significantly lower (18.2% less than our method). Note that the comparison is based on similar model size (~13B). This underperformance can be attributed to the fact that charts represent only a small fraction of scientific diagrams.
>
> **Table: Performance Comparison between ChartLlama and CoR.**
>
> |  | Bio. | Chem. | Geo. | Math | Physics | Arch. | CS   | Elec. | Energy | Mater. | ME   | total |
> |------------|------|-------|------|------|---------|-------|------|-------|--------|--------|------|-------|
> | ChartLlama | 35.6 |  15.3 | 32.5 | 22.4 |    30.6 |  31.6 | 22.4 |  27.6 |   29.4 |   23.9 | 28.0 |  27.2 |
> | CoR | 42.0 |  34.6 | 51.8 | 51.4 | 52.8 |  45.1 | 43.3 |  33.5 |   48.7 |   44.9 | 51.9 |  45.4 |
>
> **Reference**
>
> [4] Han, Y., Zhang, C., Chen, X., Yang, X., Wang, Z., Yu, G., ... & Zhang, H. (2023). Chartllama: A multimodal llm for chart understanding and generation. arXiv preprint arXiv:2311.16483.
>
> > ### Question regarding real-world scientific diagrams
>
> **Original Comments**:
>
> *"How does the proposed method fare on parsing out real-world scientific diagrams?"*
>
> **Response**:
>
> The Massive Multi-discipline Multimodal (MMMU) dataset (Yue et al., 2024a), which serves as the primary dataset for our experiments, is a real-world collection featuring a diverse range of multimodal questions derived from college exams, quizzes, and textbooks.
>
>
> > ### Computational complexity comparison
>
> **Original Comments**:
>
> *"Do you have any hint of comparisons about those computational complexities and how they scale?"*
>
> **Response**:
>
> Great question! We present the average handling times in generating region candidates for SAM2 and the OpenCV module within the CoR framework in the table below. Images are categorized into three groups based on their total pixel count: small (<250K), medium (250K–1M), and large (>1M). The results demonstrate that CoR significantly outperforms the network-based solution, SAM2, in terms of efficiency.
>
> ---
>
> **Table: Computational Efficiency Comparison in (averaged) seconds.**
>
> | Model  | Small (<250K) | Medium (250K–1M) | Large (>1M) |
> | -------- | -------- | -------- | -------- |
> | CoR (cpu)| 0.58 | 1.23   | 2.11 |
> | SAM2 (A100)| 24.49 | 24.28 | 25.66  |
>
> ---
>
>
> > ### More Discussion on Limitations Regarding Performance Boundary
>
> **Original Comments**:
>
> *"I want to hear about the real limits of this approach."*
>
> **Response**:
>
> That’s a fair concern! Currently, the CoR method excels at interpreting diagrams composed of simple geometric shapes, such as bar plots, relationship graphs, flowcharts, and tables. However, it encounters challenges with certain scientific diagrams that combine natural imagery with geometric elements, such as topographic maps and 3D model illustrations. We plan to include these failure cases in the final draft to better delineate the limitations of our approach and provide a more comprehensive understanding of its boundaries.
>
> > ### Budget parameter B in Step 3 (Region Merging)
>
> **Original Comments**:
>
> *"Discussing at least the trade-off between a large and a small B would be nice. "*
>
> **Response:**
>
> Thank you for the comment! The budget parameter B indeed plays a critical role in the Region Merging step, and we appreciate the opportunity to discuss this  further. The choice of B requires a careful balance, taking into account multiple factors such as the specific usage scenario, image complexity, desired system response time, and the VLM's capacity in handling long contexts.
>
> * **Trade-offs**: a) A large budget B allows for finer granularity by analyzing a greater number of regions. While this can provide more detailed semantic information, it comes with trade-offs, including increased computation time and longer context sequences for the VLM to process. Interestingly, as shown in the sensitivity analysis (Figure 4), a larger budget does not always lead to improved performance. This is likely because excessively long contexts may overwhelm the model, introducing noise and distractions. b) Conversely, a small budget is computationally efficient and results in shorter contexts but may fail to capture all relevant details, particularly for complex images. Thus, the budget needs to be chosen thoughtfully to balance these competing priorities.
>
> * **Our Choice of B = 5**: In our experiments, we selected B = 5 as a heuristic choice, which we found to perform well across various tasks. However, we acknowledge that this is not universally optimal and that dynamically adapting B based on image characteristics and downstream tasks is an open problem worthy of further investigation.

---

### Author Response · Authors · 2024-11-20
**Summary of response -- thanks to all reviewers for the thorough and insightful feedback**

We thank all reviewers for their constructive and valuable feedback!

We are honored that the reviewers recognize the novelty and practicality of our approach in combining classical computer vision techniques with modern VLMs (R1, R2, R3), inspiring future research (R1). Reviewers also appreciated the plug-and-play and white-box nature of our method (R1, R4), along with the cost-effectiveness and efficiency of the CoR method using CPU-based OpenCV components (R1). We are also grateful for the recognition of our thorough experimental design and extensive benchmarking of prompting strategies (R2, R3), which support our solid results and notable performance gains on state-of-the-art models like GPT-4o (R1, R4). Lastly, we appreciate the reviewers’ positive feedback on the clarity and presentation of our paper, noting its well-written structure, clear figures, and straightforward narrative that aid comprehension (R2, R4).


We have addressed the reviewers’ comments and concerns in individual responses to each reviewer.

(\* As abbreviations, we refer to **Reviewer sLjr** as R1, **Reviewer WY7p** as R2, **Reviewer dTRw** as R3, and **Reviewer fCtq** as R4 respectively.)

---

### Public Comment · ~Seunghyuk_Cho1 · 2025-03-25
**Plans for releasing the code of chain-of-region**

Thanks for publishing a great work about diagram understanding with tool assistants! Currently, I'm really inspired by the work so I want to try to reproduce the results in the paper. Do you have any plan to make the code used to run the experiments of the paper public or share to others?

---

> ### Public Comment · ~Xue_Li15 · 2025-04-01
>
> Thank you for your interest in our work on the CoR paper. However, due to intellectual property constraints and company policy, we are unable to share the codebase at this time.

---

### Meta-Review · Area_Chair_gWC1 · 2024-12-20

**Metareview:**

This paper proposes "Chain-of-Regions" (CoR), an approach to improve the ability of Visual Language Models (VLMs) to understand scientific diagrams. CoR uses traditional computer vision techniques to break down images into smaller regions and then feeding them to the VLM. The reviewers unanimously argue for the acceptance of the paper. Reviewers think the work is novel which introduces a new approach that combines traditional CV techniques with modern VLMs, offering a fresh perspective on diagram understanding. For example, reviewer sLjr mentioned that "The novel and refreshing contribution of this paper is that employing old computer vision techniques in a smart way can be very helpful with modern visual language models." The experiments also validated the effectiveness of the approach that shows significant performance gains on various task, and the approach can run CPUs, making it cost-effective. Meanwhile, the reviewers raised questions about generalizability and heuristics-based nature of the approach.

**Additional Comments On Reviewer Discussion:**

The reviewers asked a long list of questions that are valuable for strengthening the contribution of the work, such as theoretical justification, parameter sensitivity, and other technical details. The authors addressed these questions well with clarification and additional experiments.

---

### Decision · Program_Chairs · 2025-01-22

Accept (Poster)